# Is There a Role for Medication in Managing Delirium with Dementia?

**DOI:** 10.3390/geriatrics7050114

**Published:** 2022-10-07

**Authors:** Elizabeth L. Sampson, Frederick Graham, Andrew Teodorczuk

**Affiliations:** 1Department of Psychological Medicine, East London NHS Foundation Trust-Royal London Hospital, London E1 1BU, UK; 2Division of Psychiatry, University College London, London W1T 7NF, UK; 3Division of Medicine, Princess Alexandra Hospital, Woolloongabba, QLD 4102, Australia; 4School of Nursing, Queensland University of Technology, Brisbane City, QLD 4000, Australia; 5Centre for Health Services Research, Faculty of Medicine, The University of Queensland, Herston, QLD 4006, Australia; 6Metro North Mental Health Service, The Prince Charles Hospital, Brisbane, QLD 4032, Australia; 7School of Medicine and Dentistry, Griffith University, Southport, QLD 4222, Australia; 8Faculty of Medicine, The University of Queensland, Brisbane, QLD 4006, Australia

**Keywords:** delirium, delirium superimposed on dementia, dementia, inappropriate prescribing, nursing care, organizational culture, older adults, psychomotor agitation, psychosocial intervention

## Abstract

People with dementia are more likely to develop delirium. We conducted a brief literature search and give a pragmatic overview of the key issues. Making rational and safe prescribing decisions is highly influenced by organisational culture and embedded staff practices. Comprehensive assessment for unmet physical, psychological, and social needs is an important intervention in itself. Taking a broad overview of possible pharmacological interventions should include stopping inappropriate medications and prescribing for key drivers of the underlying causes of delirium. Prescribing psychotropic medications may be indicated where there is significant distress or risk to the person with dementia and risk to those around them. It is vital to consider the dementia subtype and, where possible, involve family and friend carers in the decision-making process. Medications should be prescribed at the lowest possible dose for the least amount of time after carefully weighing risks versus benefits and documenting these. While these cases are challenging for staff and families, it can be rewarding to improve the quality of life and lessen distress for the person with dementia. There are also opportunities for informing family and friend carers, educating the wider multidisciplinary team, and promoting organisational change.

## 1. Introduction

“Albert is a 79-year-old man, admitted to a general hospital ward for older people with community-acquired pneumonia, early signs of sepsis, and requires intravenous fluids and antibiotics. He was diagnosed with Alzheimer’s disease in the local memory clinic three years ago and is now in the moderately severe stage, with a recent Addenbrookes Cognitive Exam (revised) score of 55. He lives at home with his wife and needs increasing support with his activities of daily living, such as dressing and using the bathroom. He has a past medical history of chronic kidney disease and myocardial infarction and still smokes 20 cigarettes a day. You are a junior doctor or resident on duty overnight and are called to the ward at 3 am by one of the nurses. Albert is described as “suddenly agitated and aggressive”; he is walking around, trying to get out of the door, and shouting. He is shaking other patients to wake them up and has also pushed a nurse. You arrive and are asked to “do something”. How do you proceed?”

People with dementia and other neurodegenerative disease are much more likely to become delirious. Nearly half of people with dementia will experience an episode of delirium during hospital admission [1]. Delirium superimposed on dementia (DSD) is also common in care home residents [2]. Cognitive frailty is frequently associated with physical frailty, and people with DSD often have multiple co-morbidities, inevitably associated with complex polypharmacy. They are likely to be prescribed multiple medications with both unintended and intended psychotropic properties [1]. Not surprisingly, they are in a state of high cognitive vulnerability, and clear stressors for delirium may not be identified. Moreover, persistent delirium in someone who is frail may accelerate the progression of underlying early dementia and lead to a downward, and hard to reverse, spiral of mental and physical decline. Conversely, persistent delirium is often missed where it is just assumed that a “progression” of dementia has occurred [3]. DSD, especially in the context of increasing frailty, may indicate that the person is nearing the end of life [4].

While nonpharmacological approaches remain the mainstay of delirium prevention and management, sometimes medication is necessary. Sub-optimal prescribing may have a profound impact on this vulnerable group of patients, family and friends carers, and and staff. It is extremely challenging to research delirium in people with dementia, and the complexities of conducting trials of pharmacological interventions mean we have sparse evidence on which to base prescribing decisions [5,6]. In this article, we take a pragmatic, broad, and practice-based approach to pharmacological interventions for DSD. 

## 2. Rational Prescribing for People with Delirium on Dementia

### 2.1. The Challenge of Balancing Risks and Benefits

As doctors and healthcare practitioners who prescribe, we are often called to see people with DSD because they are experiencing changed behaviours and psychological distress. While many people living with dementia may already have behavioural and psychological symptoms of dementia (BPSD), delirium may increase the prevalence and severity of BPSD two to three-fold [7,8], especially agitation and aggression [9,10]. In such instances, we can experience multiple situational pressures to prescribe psychotropic medications [11]. Often there is a sense of urgency, with a perception that safety is becoming compromised. This is understandable, as the odds of occupational violence toward staff in general hospital wards increases eleven-fold in patients with delirium and seven-fold in those with dementia [12]. Disruptions to clinical workflows and staffing resources may add to this pressure. We argue that while medication may have a role in some instances where harm from violence is carefully weighed against harms associated with medication side-effects, such prescribing must always include awareness of the clinical presentation, its context, and the workplace and organisational factors contributing to situational pressures. 

### 2.2. Understanding and Assessing Behaviour as Unmet Needs

At face value, behaviours described as “agitation” and “aggression” indicate high risk, both for the person with DSD and staff caring for them. However, clinicians’ initial descriptions of behaviour may prove problematic and misleading. The term “agitation”, for example, is often used imprecisely and with limited contextual descriptors, becoming a catch-all for behaviours regarded as “problematic or disruptive” [13]. Agitation should be considered an indicator of distress secondary to unmet needs [14,15]. 

Arguably, the term “aggression” is similarly problematic. While aggression may occur as a reaction to receiving unwanted care (i.e., “resistance to care”), requiring a change of approach and not sedation, if the context of aggression is left unexplained or unexplored it may be interpreted by staff as “unprovoked” behaviour requiring sedation. Describing distress behaviours as “aggressive” may be directed towards a call for rapid intervention that bypasses sufficient assessment [13]; “agitation or aggression” may become code or a signal for injudicious prescribing or use of restraint. Physical restraint, i.e., any method that reduces the ability of the person to move freely, is used with widespread variability across different countries and settings. For example, in intensive care units, physical restraint use varies widely from 0% in some European countries to more than 75% in North America [16].

Therefore, comprehensive assessment is a crucial intervention in itself [17], enabling the identification of physiological precipitants like constipation and infection or psychological needs like boredom, overstimulation [18,19], and the absence of familiar people. It follows that the first step in establishing safe and effective interventions (with or without medications) is to identify and assess thoroughly the true nature of acute behavioural change.

### 2.3. The Importance of Organisational Context

In many general hospital wards, institutional drivers around routines, efficiency, and risk reduction often limit the capacity of staff to provide person-centred dementia and delirium care in response to symptoms of distress [20,21,22,23]. To achieve expected organisational outcomes with scarce resources in time-poor settings, staff must divide their limited time to try and benefit the maximum number of patients. As such, they are enculturated into a task-based utilitarian approach to care, becoming inflexibly committed to routines, rules, duty, responsibility, compliance targets, streamlining, and “being in charge”. These attributes tend to then trigger aggressive and resistant behaviours in those they care for, a patient cohort that requires a more flexible, less rigid approach to care based on mutualism and facilitation of patient autonomy and identity [24]. In a recent ethnography of hospital wards, inflexibly timetabled routines of care were found to trigger patterns of resistance in patients with dementia resulting in staff burnout [25]. Furthermore, ongoing failures to “manage aggression” can lead to self-protective behaviours in nurses, which ultimately limit therapeutic person-centred interactions with patients [26]. Nurses may develop a “socialised care futility”, believing that therapeutic, responsive behaviour management is no longer possible [27].

These learned responses are more likely in organisations that do not value dementia care, where staff receive inadequate training and may be “educationally starved”, and the physical environment is inappropriate for people with dementia [22]. When organisational priorities support staff in dementia care, responses to “agitation and aggression” are more likely to be swift, proactive, and investigative. Some organisations have prioritised specialised dementia care units [28,29] and shared care wards [30,31]. Although more studies are needed, these innovations demonstrate that when the care setting is reorganised for people with dementia, quality dementia care can be achieved. However, for most hospital settings, a general absence of research, training, and established models of how best to care for people with dementia [31] means that nurses are mostly left to rely on their collective experience to provide care [17]. Unfortunately, relying on collective experience alone may lead to the embedding of less-than-ideal responses to patients in distress. A recent simulation study of nurses revealed that the predominant and learned workplace practice for managing agitation was quick administration of antipsychotics without first assessing for unmet needs like pain [32].

It is within this overall context that prescribers are often asked to provide “urgent” pharmacological solutions to challenging and complex behavioural situations. While many of the aforementioned factors are not controllable by prescribers, they must develop a situational awareness, carefully considering how ward routine, design, and staff interactions may directly influence the presentation of behavioural distress in a specific case. With comprehensive assessment alongside an astute situational appraisal (however brief), a clearer picture of what is achievable in the immediate circumstance may become apparent. However, there are times when the confluence of imperfect environmental, organisational, and staffing factors means that the only safe course of action remaining may be to prescribe a sedating medication.

### 2.4. Working as a Multidisciplinary Team

Involving the full multidisciplinary team (MDT) in managing behaviour in DSD is essential. Although full information may not be available “out of hours”, valuable person-centred knowledge, considered cumulatively, can powerfully contribute to an effective person-centred care plan. Professional hierarchy may mean that some care staff, such as personal care workers and nursing assistants, who work through nights and weekends, lack the opportunity to share critical clinical information derived from direct care experiences [11]. The MDT should address modifiable risk factors as a priority, including hearing aids and glasses, promoting exercise and mobilisation, and ensuring catheters are removed. The introduction of psychosocial approaches such as planned activity and quiet times alongside simulated family presence through technology could all potentially alleviate distress in these cases without the need for medication [33]. Where possible, nonpharmacological approaches should involve family and friend carers in understanding how to provide assurance and orientation as well as with interpreting the various unmet physical and psychological needs the person with DSD may be attempting to communicate. Thus, family and friend carers can be valuable partners with the clinical team, supporting the exchange of information about the person with DSD, which further enhances individualised care. 

Too often, prescribers review patients without involving the staff responsible for developing and providing nonpharmacological care. For example, a social worker may be aware of biographical information they had not thought important enough to share, but when participating in problem-solving behaviour management, it is revealed as critical information (e.g., past trauma or sensitive lifelong habits). Well-timed analgesia may mean that recreational activity and hygiene care are not resisted, or alternatively, antipsychotic medication may have a temporary role in reducing paranoid delusions allowing engagement in social, recreational, and personal care activities. At other times, medications may cloud a patient’s awareness, contributing to combative reactions to attempts at engaging them in activity, requiring downward titration. When prescribing closely supports and responds to the daily progress of planned nonpharmacological care, effective and safe medication is likely. In addition, medications will be better targeted, responsive, and safer and the whole team (including family and friend carers) is more likely to succeed in achieving effective therapeutic care and developing relational expertise through the process.

### 2.5. Making Safe Prescribing Decisions

In the absence of “gold standard” randomised controlled trial (RCT) evidence and meta-analyses, how can we ensure prescribing is rational and safe? Most national delirium management guidelines take a pragmatic approach and acknowledge that in some circumstances prescribing antipsychotics, which may not be licensed for the prevention or management of delirium, may be necessary [34,35,36]. Hospitals may adapt existing national guidance to tailor their own policies. Prescribing within these frameworks can support decision-making and, in some countries, may be done without the person’s informed consent, underpinned by national or local legal frameworks. For example, in England, the Mental Capacity Act (2005) allows prescribing for people who lack the mental capacity to consent for this if it is in their “best interests”. Briefly documenting the legal framework used to justify prescribing in DSD is essential and may also safeguard the prescriber. This will vary across jurisdictions. For example, in Australia, this may be written as “an intramuscular injection of an antipsychotic was given under section 63 of the guardianship act”.

## 3. What Drugs May Be Beneficial for People with Delirium on Dementia?

### 3.1. Drugs Which May Help Address Potential Underlying Causes of Delirium

Prescribing in DSD often narrowly focuses on psychotropic medications, principally sedatives and antipsychotics. Some situational drivers for this are described earlier. It is important to take a broad and holistic stance and not become “diagnostically overshadowed”. Taking an approach that optimises cerebral function, using a holistic and person-centred framework such as Comprehensive Geriatric Assessment can be a powerful intervention in itself. Even a brief comprehensive assessment during an emergency may highlight other prescribing options. For example, managing hypoxia, preventing venous thromboembolism (VTE), particularly in hypoactive delirium, and promoting mobilisation by prescribing analgesics are all important “prescribing” interventions for delirium. Reviewing the medication chart and de-prescribing potentially harmful medications is also a key step; using Beers criteria provides an evidence-based framework [37]. Taking a holistic “assessment as intervention” approach widens the range of potentially helpful prescribing options for managing DSD. Clinical mnemonics that act as heuristics (rules of thumb) to prompt structured assessment of DSD, for example, “TIME” (Think, Investigate, Manage, Engage) [35] or “PINCHME” provide a structure for this, bearing in mind that there is often no single “cause” for delirium (Box 1) [38].

Box 1Examples of prescribing interventions for DSD.
Pain: this can be difficult to detect in people with dementia, requiring careful assessment of changes in behaviour, facial expression, and function [39]. Prescription of regular analgesics, i.e., paracetamol, may decrease the length and severity of delirium, but care is needed with opioids and other drugs with psychoactive properties [40].Infection: urinary tract infections are often overestimated as a cause of delirium in people with dementia [41], and it is easy to overlook dental infections [42] or atypical presentations, for example, COVID 19, where delirium can be a presenting symptom in the absence of classic respiratory signs [43].Nutrition: support with eating and drinking is essential in people with DSD. Adequate nutrition is a mainstay of effective multicomponent delirium interventions such as the Hospital Elder Life Programme (HELP) [44]. Assessment by a dietician and prescription of nutritional supplements should be considered.Constipation: checking the bowel chart is a commonly missed step, but managing this may be a key prescribing intervention.Hydration: oral fluids should be supported, but in people with more severe dehydration and electrolyte imbalance, temporary use of intravenous fluids may be indicated.Medication: de-prescribing drugs with anticholinergic properties may support cognitive function. Various online tools grade drugs on their anticholinergic effects, i.e., MEDICHEC (https://medichec.com/) (acessed on 4 October 2022). Other medications commonly prescribed in older people that may cause acute confusion and either mimic or precipitate delirium include: levetiracetam, gabapentin, benzodiazepines, and antidepressants [45].Environment: consider the environment and how the person with DSD interacts with this. Do they have their glasses and hearing aids? Is it too noisy and stimulating? Can modifications be made, for example, moving to a quieter room or improving lighting?


Deprescribing is a useful approach; although challenging and perhaps less immediately helpful in acute situations, it may bring long-term benefits. Guidance such as the STOPP/START criteria highlights drugs that may be inappropriate or have been omitted. MDT input, particularly pharmacy, is vital in making complex decisions where the lowest benefit-to-harm ratio and the lowest likelihood of withdrawal or rebound syndromes need to be considered [46]. Again, careful consideration and evaluation of the planned nonpharmacological care is a must. There are some specific issues in DSD. While there is little evidence that cholinesterase inhibitors are useful for treating delirium [47], accidentally omitting them from a drug chart when someone changes care setting or the person is too unwell to take orally may precipitate or worsen delirium [48]. The use of alternative routes of administration, i.e., cholinesterase inhibitor patches, may support central cholinergic activity. Suddenly stopping a range of psychotropic medications, including selective serotonin reuptake inhibitors (SSRIs), benzodiazepines, and opioid medications may precipitate discontinuation or withdrawal reactions that can mimic or precipitate delirium. Nicotine replacement therapy should also be considered when a person with DSD cannot smoke. Trying to leave the care setting may be due to deeply ingrained patterns of smoking habits. 

### 3.2. When, What, and How to Prescribe

In certain situations, on an individual case-by-case basis and as a last resort, rational prescribing to manage symptoms of delirium superimposed on dementia may be necessary. Ideally, this should only be undertaken following the implementation and potential failure of nonpharmacological approaches. 

So, what is rational prescribing? In essence, it is a judicious last resort prescribing when there is significant distress or risk to the patient or others, typically due to hyperactive delirium [34]. It should involve starting low-dose medication following the seven principles of safe prescribing in older people (Box 2) and reviewing daily for improvements [49].

Box 2The seven principles of safe prescribing in older people.
Ensure benefits outweigh risk.Involve patients and families in treatment decisions.Ensure medication plan is part of treatment plan.Start with a low dose and increase gradually.Manage adverse effects appropriately.Avoid unnecessary prescribing.Withdraw gradually.


In clinical settings, antipsychotic medications may be advocated at a low dose. However, these should be prescribed within the understanding that, in general, the evidence base for antipsychotics shows that at a population level, they are ineffective treatment for delirium. This has been demonstrated in a systematic review by Neufeld and colleagues [50]. Moreover, it appears that the more robust the study, the less favourable the outcomes are for prescribing antipsychotic medications. A well-conducted RCT study undertaken in palliative care patients by Agar and colleagues [51] showed that antipsychotic medications are worse than placebo in terms of side effects and led to worsening delirium symptoms. It could be argued that antipsychotics may be effective in targeting specific psychotic symptoms secondary to delirium, for example, severe delusions or hallucinations; however, there is currently very little robust evidence to support this approach. 

Moreover, antipsychotic medication is associated with well-known side effects. These include worsening cognitive dysfunction, parkinsonism, akathisia, postural hypotension, dystonia, anticholinergic effects, sedation, and tardive dyskinesia. In addition, longer-term antipsychotics increase the risk of stroke [52]. 

Within the paradigm of rational prescribing, prescribers should be clear about what the purpose of prescribing is, which is usually with the aim of reducing risks or distress. Equally, though, one must remain mindful of the fact that we could be sedating and rendering someone from a hyperactive delirium into a hypoactive delirium, thereby ultimately not “treating” the delirium at all. As far as possible, psychotropic medications should be prescribed as single agents only, avoiding layering on of multiple agents via different routes (oral and by injection) and different administration frequencies, for example, using regular and “as required” medications simultaneously. Rationalising multiple psychotropic medications into a single agent, given at low dose and regularly, may enable optimisation of clinical and psychosocial care. Clear documentation in the notes concerning what the target symptoms are, and when antipsychotics or other medications should be discontinued is critical. Unfortunately, antipsychotics are frequently not deprescribed when delirium has subsided, and therefore, this is a real risk in patients with DSD [53].

Melatonin, despite not being widely licensed for this purpose, is sometimes prescribed for treatment or prophylaxis of delirium. Melatonin has been shown in one RCT to have no difference on mortality or length of stay but reduces the duration of delirium [54]. Other RCTs have shown that melatonin did not prevent delirium in patients undergoing cardiac [55] or orthopaedic surgery [56], or in intensive care units [57]. In general, benzodiazepines should not be prescribed for delirium. They have been shown to worsen outcomes and can lead to a paradoxical worsening of behaviour. However, these are correctly indicated in delirium secondary to alcohol dependence and withdrawal. In addition, emerging evidence appears to suggest that it is important to differentiate delirium from catatonic symptoms If there is a clear catatonic phenotype, then there may be a place for benzodiazepines. This should be specifically clarified, as antipsychotic medications may make catatonia worse.

In addition, specific care should be taken with patients who may have Lewy body dementia (DLB) and superimposed delirium. In this highly vulnerable group, antipsychotics should be used with great caution due to patients’ profound sensitivity to these drugs. Hence, if prescribing in DSD, it is important to ensure that the subtype of dementia is clearly differentiated. This can be challenging when mental health, primary care, and acute hospital notes remain separate in many health systems and vague diagnoses such as “cognitively impaired” or “dementia” are the only information documented. This is particularly important due to the phenotypic overlap of persistent delirium and DLB, where both conditions involve fluctuations and hallucinations [58].

In summary, rational prescribing in DSD is complex, potentially problematic and should only be undertaken as a last resort. An accurate understanding of the dementia subtype is vital before reaching for drugs, in addition to clearly demarcating target symptoms and when to cease prescribing. Moreover, if antipsychotics are prescribed concerted efforts must be undertaken to stop treatments once delirium has resolved or it is deemed antipsychotics are ineffective and/or indeed if nonpharmacological care has become more effective. Side effects should also be actively sought as the risk of doing more harm than good is high in this vulnerable frail patient group. 

## 4. Involving Family and Friend Carers

As mentioned throughout this article, actively involving patients’ families or friend carers in managing delirium is essential for success. They should be involved, if possible, in decision-making processes with regard to the limitations of the prescribing approach, risk, and potential benefits. However, there must be recognition that, in general, public awareness about delirium is low. Most patients’ families will never have heard of delirium. Therefore, witnessing a loved one experiencing delirium can be extremely distressing for the family and the patient. To address this, it is essential to educate the family clearly and support this learning with patient information materials (that are patient centred and understandable) concerning why their relative has changed suddenly (Box 3).

Box 3Useful resources for family and friend carers.
Dementia Australia factsheet “Delirium and dementia”: https://www.dementia.org.au/sites/default/files/helpsheets/Helpsheet-DementiaQandA21_Delirium_english.pdf (accessed on 4 October 2022)Dementia UK: “Delirium (confusion)”: https://www.dementiauk.org/delirium/ (accessed on 4 October 2022)UK Alzheimer Society: “Delirium—symptoms, diagnosis and treatment”: https://www.alzheimers.org.uk/get-support/daily-living/delirium (accessed on 4 October 2022)Australasian Delirium Society “Community Information”: https://www.delirium.org.au/Community-Information (accessed on 4 October 2022)Delirium Public Awareness- animation: https://www.youtube.com/watch?v=BPfZgBmcQB8 (accessed on 4 October 2022)


This is especially important as a patient with DSD may be nearing the end of their life, and sharing a clear understanding of processes and expectations will impact the grieving process that may follow. Points to emphasise are that if a person acts out of character, it is the illness driving the behaviour and not the person. In addition, emphasising the role of nonpharmacological approaches is key, especially in determining how to normalise and personalise care. Managing expectations and sharing information that a person with DSD may be at the end of life is essential. Pivoting towards an understanding that the goal may not be to prolong life but to ensure that what life is left is free of distress and should involve spending meaningful time with families is important. 

## 5. Conclusions

Returning to Albert, aged 79, admitted to an acute hospital ward in a state of acute behavioural distress. How could we implement this rational prescribing approach to benefit him? His situation is complex and challenging to manage, so where do you start?

Even in the most acute situation, it is vital to take a step back and briefly survey the context and culture in which the acute disturbance is happening. Ward culture and socialised care futility amongst staff often lead to the instinctive response to request prescription of psychotropic drugs. A brief assessment, using a simple clinical heuristic or rule of thumb such as “PINCHME” (Box 1), and clinical examination, where possible, informed by available information from those that have the most direct experience of caring for the patient, may be a powerful intervention. For example, the family or friend carer or nursing assistant may provide vital information to support initial nonpharmacological interventions. If these are not possible, such information may still inform prescribing, for example, the need for antibiotics for a chest infection, nicotine replacement therapy, or laxatives. If the patient or others are at a significant risk of harm or the patient is highly distressed, then it may be reasonable to prescribe medications to manage this, taking into account the patient’s co-morbidities, type of dementia, and the risk:benefit ratio. Importantly, the medication should be reviewed and, if possible, stopped. 

After the acute event and when family and other key members of the MDT are available, opportunities to personalise care and tackle issues to do with dignity, spiritual needs, and reducing suffering can be aligned to prescribing goals. Amongst all this complexity, it is important to see this kind of case as an opportunity. Rational prescribing and deprescribing is a framework, firstly through which to “do no harm” and reduce the distress felt by the patient, the staff trying to care for them and their family or friend carers, and secondly, a way to inform, educate and strengthen MDT working and organisational culture. 

## Data Availability

Not applicable.

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
