# Peer review of "Is There a Role for Medication in Managing Delirium with Dementia?"

_geriatrics, 2022, doi:10.3390/geriatrics7050114_

Round 1

Reviewer 1 Report

The authors present a much needed pragmatic approach to a difficult and common clinical problem. They highlight the lack of good evidence but make very sound suggestions for management.  They are able to articulate important principles relevant to this issue.  I was especially appreciative of the need to define "aggressive" and "agitation" with respect to the context in which these behaviors are observed.  This is very sound advice which is often not thought about.

Author Response

Thank you very much for your supportive comments. 

Reviewer 2 Report

I’d like to commend the authors for their thoughtful review on the approach to the “behavioural and psychological symptoms” of delirium. This is very well written and resonates with my experiences as a clinician in an acute medical setting. The authors strike a nice balance between honoring the patient and acknowledging the exigencies of care in the context of (often intransigent) hospital culture. I offer the following comments, which I hope will strengthen this piece even further.

Major points:

1)      I would suggest the authors to consider broadening their statement of when to consider medications for the management of delirium’s neuropsychiatric symptoms. Under section 3.2, they write that it may be necessary “when there is significant distress or risk to the patient, typically as a consequence of hyperactive delirium.” Would the authors also consider prescribing for psychotic symptoms or to facilitate indicated treatment (e.g., prevent risk of self-extubation)? Also, considering their statement under section 2.2, would the authors consider medication if the patient represented a risk to other patients—that is, not just to themselves or to carers?

2)      Under section 2.5, the authors suggest “briefly documenting the legal framework used to justify prescribing in DSD.” Can they elaborate on what they mean here? Would this involve citing a statute in the prescriber’s country, jurisdiction, hospital policy, etc.? Can the authors provide some concrete suggestions of what this looks like?

3)      In section 3.2 in the paragraph following Box 2, the authors write that antipsychotics are “ineffective treatment for delirium.” Whereas I agree with this statement, I would ask the authors to consider differentiating the treatment of delirium’s core features from treating its underlying cause(s) and treating its “behavioural and psychological”/neuropsychiatric symptoms. It is true that antipsychotics don’t reverse delirium (the core syndrome) or cure its underlying causes (such as sepsis); however, whether they may improve neuropsychiatric symptoms has not been studied rigorously.

4)      As for specific agents, I would caution against suggesting melatonin. Recent, robust RCTs have failed to find melatonin effective at even preventing delirium (much less, treating its neuropsychiatric symptoms): Ford JAGS 2020 (Health Heart-Mind trial), Oh AJGP 2021 (RECOVER study), Wibrow ICM 2022 (Pro-MEDIC study). Furthermore, melatonin is not regulated in every country.

5)      The statement that antipsychotics “must never be prescribed” in DLB may overstate the case. True, neuroleptic sensitivity is a key feature of DLB, but agents with limited D2 antagonism may occasionally be used successfully in DLB (e.g., quetiapine or clozapine). The recently approved pimavanserin could also be considered as it lacks clinically relevant D2 antagonism.

Minor points:

1)      Please use either “antipsychotic” or “neuroleptic,” rather than alternating between them.

2)      In section 3.1, please correct “Beer’s” to “Beers,” as this was named after the geriatrician Mark Beers.

3)      In the end of section 3.1, suggest changing “precipitate withdrawal reactions” to “precipitate discontinuation or withdrawal reactions” as SSRI discontinuation is not properly a withdrawal syndrome.

Author Response

Dear Reviewer 2,

Thank you for your positive review and suggestions on how to strengthen the paper further. We have responded to all the points, as described below,

Yours sincerely,

Elizabeth Sampson

MAJOR POINTS

1. Comment: I would suggest the authors to consider broadening their statement of when to consider medications for the management of delirium’s neuropsychiatric symptoms. Under section 3.2, they write that it may be necessary “when there is significant distress or risk to the patient, typically as a consequence of hyperactive delirium.” Would the authors also consider prescribing for psychotic symptoms?

Response: We have added more detail to section 3.2 (please see our response to reviewer’s comment number 3).

Comment: (Use of antipsychotics)...to facilitate indicated treatment (e.g., prevent risk of self-extubation)?

Response: We understand that approaches vary across different countries and settings. In this circumstance we would suggest the use of ICU sedation protocols and appropriate sedation, rather than treating the distress of awareness of intubation with an antipsychotic. We have thereof not made any changes to the text regarding this point.

Comment: Also, considering their statement under section 2.2, would the authors consider medication if the patient represented a risk to other patients—that is, not just to themselves or to carers?

Response: We have acknowledged there may be risks to others and added this in section 3.2 (paragraph 2)

2. Comment:  Under section 2.5, the authors suggest “briefly documenting the legal framework used to justify prescribing in DSD.” Can they elaborate on what they mean here? Would this involve citing a statute in the prescriber’s country, jurisdiction, hospital policy, etc.? Can the authors provide some concrete suggestions of what this looks like?

Response: We agree this comment may appear a little abstract but it is difficult to explain, given national, state and federal variations across different countries. We have added a brief illustration of what this could look like in one jurisdiction. “This will vary across jurisdictions. For example, in Australia this may be formatted as ““an intramuscular injection of an antipsychotic was given under section 63 of the guardianship act”.

3. Comment: In section 3.2 in the paragraph following Box 2, the authors write that antipsychotics are “ineffective treatment for delirium.” Whereas I agree with this statement, I would ask the authors to consider differentiating the treatment of delirium’s core features from treating its underlying cause(s) and treating its “behavioural and psychological”/neuropsychiatric symptoms. It is true that antipsychotics don’t reverse delirium (the core syndrome) or cure its underlying causes (such as sepsis); however, whether they may improve neuropsychiatric symptoms has not been studied rigorously.

Response: We agree with the reviewer that antipsychotics could in theory benefit specific psychotic symptoms and have added some text to broaden the argument. “It could be argued that antipsychotics may be effective in targeting specific psychotic symptoms secondary to delirium, for example severe delusions or hallucinations, however, there is currently very little robust evidence to support this approach.”

4. Comment: As for specific agents, I would caution against suggesting melatonin. Recent, robust RCTs have failed to find melatonin effective at even preventing delirium (much less, treating its neuropsychiatric symptoms): Ford JAGS 2020 (Health Heart-Mind trial), Oh AJGP 2021 (RECOVER study), Wibrow ICM 2022 (Pro-MEDIC study). Furthermore, melatonin is not regulated in every country.

Response: We have amended this section to include a more nuanced discussion on melatonin, using references suggested by the reviewer. “Melatonin, despite not being widely for this purpose, is sometimes prescribed for treatment or prophylaxis of delirium. Melatonin has been shown in one RCT to have no difference on mortality or length of stay but reduced the duration of delirium (de Jonghe, van Munster et al. 2014). Other RCTs have shown that melatonin did not prevent delirium in patients undergoing cardiac (Ford, Flicker et al. 2020) or orthopaedic surgery (Oh, Leoutsakos et al. 2021), or in intensive care units (Wibrow, Martinez et al. 2022).”

5. Comment:  The statement that antipsychotics “must never be prescribed” in DLB may overstate the case. True, neuroleptic sensitivity is a key feature of DLB, but agents with limited D2 antagonism may occasionally be used successfully in DLB (e.g., quetiapine or clozapine). The recently approved pimavanserin could also be considered as it lacks clinically relevant D2 antagonism.

Response: We agree that we were too absolute with this statement and have reworded this section which now reads: “In this highly vulnerable group, antipsychotics should be used with great caution due to patients’ profound sensitivity to these drugs”.

MINOR POINTS:

1. Comment: Please use either “antipsychotic” or “neuroleptic,” rather than alternating between them.

Response: We have amended the paper, so antipsychotic is used consistently and removed reference to neuroleptics

2. Comment:  In section 3.1, please correct “Beer’s” to “Beers,” as this was named after the geriatrician Mark Beers.

Response: This has been corrected

3.  Comment:  In the end of section 3.1, suggest changing “precipitate withdrawal reactions” to “precipitate discontinuation or withdrawal reactions” as SSRI discontinuation is not properly a withdrawal syndrome.

Response: This has been amended as suggested